# Review of Energy Harvesting for Buildings Based on Solar Energy and Thermal Materials

Luciana Sucupira * and João Castro-Gomes 

Centre of Materials and Building Technologies (C-MADE), Department of Civil Engineering and Architecture, University of Beira Interior, 6201-001 Covilhã, Portugal; jpcg@ubi.pt
* Correspondence: luciana.sucupira.cristino@ubi.pt

**Abstract:** Reducing the use of fossil fuels and the generation of renewable energy have become extremely important in today. A climatic emergency is being experienced and society is suffering due to a high incidence of pollutants. For these reasons, energy harvesting emerges as an essential source of renewable energy, and it benefits from the advancement in the scope of solar and thermal energy which are widely abundant and usually wasted. It is an option to obtain energy without damaging the environment. Recently, energy harvesting devices, which produce electricity, have been attracting more and more attention due to the availability of new sources of energy, such as solar, thermal, wind and mechanical. This article looks at recent developments in capturing energy from the sun. This literature review was performed on research platforms and analyzes studies on solar and thermal energy capture carried out in the last four years. The methods of capturing solar energy were divided according to how they were applied in civil engineering works. The types of experiments carried out were the most diverse, and several options for capturing solar energy were obtained. The advantages and disadvantages of each method were demonstrated, as well as the need for further studies. The results showed that the materials added to the components obtained have a lot of advantages and could be used in different energy capture types, such as photovoltaic, thermoelectric generators, pyroelectricity and thermometrical. This demonstrates that the capture of solar energy is quite viable, and greater importance should be given to it, as the number of research is still small when compared to other renewable energies.

**Keywords:** energy harvesting; solar energy; thermal materials; review

## 1. Motivation

In November 2018, the agreements between the Council of Ministers, the European Parliament and the European Commission were established in the form of new critical objectives to be followed by the European Union. Compared to 1990 levels, the new targets aim to reduce pollutant gas emissions by at least 40%. In addition, there was a greater incentive of at least 32% of renewable energies and an improvement of at least 32.5% in energy efficiency, concerning the 2007 goals. To achieve the proposed plans, collective efforts must be applied to reducing carbon and generating renewable energy, thus decarbonizing the environment [1].

The European Commission also presented the 2030 Energy and Climate Package, through which Member States should develop integrated national plans that address energy advancement and climate improvement. The idea is to contribute to a low carbon economy's progress and develop an energy system that guarantees energy availability to the entire population. The energy transition is based on complete decarbonization with electricity production from renewable sources [2].

Each Member State has a strategy to consolidate this Plan, thus aiming at a more competitive economy. The system developed in Portugal foresees that the potential of the available renewable resources, such as water, wind, biomass, geothermal and the sun,

is used. The main bet is currently solar energy, as it is a solution and a source of energy capable of continuing energy transition [3].

Considering the definition of a smart city, the European Commission has proposals to help design or retrofit buildings to reduce energy bills and the incidence of carbon dioxide ($CO_2$) and obtain the building's thermal conditioning by using thermal materials. As a result, cities will be energy efficient and oriented towards using cutting-edge technologies to meet the aims of a smart city, namely, decrease energy consumption, increase the capture of renewable energy and reduce the carbon footprint [4].

There are already several methods to capture energy sustainably. In this document, solar energy and thermal materials will be analyzed. These methods use sunlight and temperature which are non-polluting and daily accessible sources of energy. With the development of cities, which are becoming larger and larger, it is increasingly necessary to obtain solutions for their growth without affecting the environment, for example, using buildings to acquire energy. The capture of solar energy through photovoltaic and thermal materials is a means of obtaining this energy and has been studied by several researchers worldwide.

The study of energy capture in the construction sector is relevant. This article aims to investigate the production of solar energy, through temperature, by analyzing which methods were used in previous research. A systematic literature review was carried out between 2017 and 2020 in the Science Direct and MDPI databases to achieve the objective. In addition to these two databases, some authors were also considered for bibliographic review from other research sources.

This article offers a review of the literature on the capture of thermal and solar energy. The technologies with high potential for capturing solar and thermal energy are covered in different large groups: the capture of energy from thermoelectric generators, using photovoltaic cells, from thermal materials and the capture of pyroelectric energy divided according to the places where it was used in civil engineering. A survey was also carried out to determine the advantages and disadvantages presented in the analyzed articles and even the need for further future studies. The themes are related to civil engineering regarding the capture of solar energy and temperature differences and can be applied in different city locations as sources of energy generation for public lighting and building efficiency. This leads the buildings to become increasingly self-sustainable and with similar energy generation and consumption.

## 2. Methodology

This systematic research on the databases resulted in a bibliographic portfolio that covers the most relevant research on solar energy harvesting. To apply this methodology, the following phases were followed:

Initially, the keywords, "Energy harvesting" and "Solar energy", used in the research platforms were defined. The period established for the research was between 2017 and 2020, as it is a new discipline for which new technologies are essential. Articles whose title was not related to the area of interest were not considered. The search did not return many articles related to temperature as a renewable energy source in civil engineering.

Then, the abstracts were analyzed to verify whether the article was related to the desired area and to civil engineering, and whether the method of harvesting solar energy and temperature difference can be applied in different buildings and facilities as a source of energy generation for street lighting and building efficiency.

Subsequently, all complete articles were analyzed, namely regarding energy harvesting methods investigated by the authors. This article mainly focuses on the information provided by global researchers on the various applications and their suitability for each material used. Furthermore, as each publication was evaluated individually, relevant data on the methods used to generate energy present in each analyzed document were quoted. However, this articles' review will help further improve the investigated methods and provide ideas for civil engineering applications.

On the Science Direct search platform, a total of 43 articles were found and in the MDPI, 8 articles were selected [5–12], all of them fitting into the theme proposed by this research. On other platforms, 22 articles were found using the two selected keywords. In total, 73 articles were considered as a reference for this document.

## 3. Types of Energy Harvesting

When analyzing the various articles focused on different harvesting energy methods, it was observed that people are becoming aware of the harm that the incidence of $CO_2$ brings to the environment. There are currently several types of energy harvesting (electromagnetic, piezoelectric, thermoelectric, solar and pyroelectric) which are applied in various areas of civil engineering. Many of these energy sources use the sun, the earth or chemical compounds. They can also use vehicles or wind loads' mechanical energy to capture renewable energy [13,14].

In recent years, several studies have been carried out proving that energy can be harvested from the sun and high temperatures, thus contributing to value other methods of obtaining energy and improving the urban environment and reducing pollutants' incidence. Considering recent developments in the technology for capturing solar energy, it is essential to identify how each type of model works: thermoelectric generator (TEG), photovoltaic cells, thermoelectric materials and pyroelectric (Figure 1).

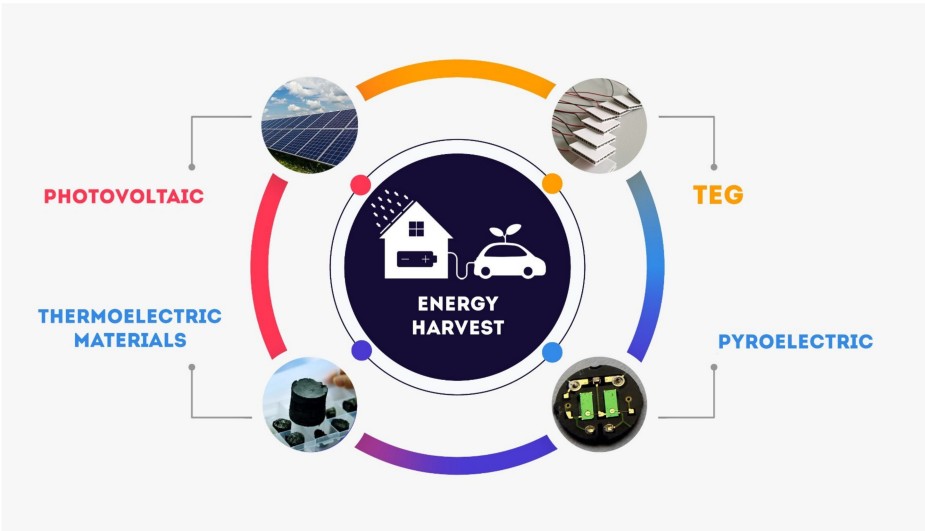

**Figure 1.** Types of solar energy harvesting addressed in this research [own authorship].

TEG works with the electrical voltage performed by two heterogeneous semiconductors joined at one end. This is due to the temperature difference, where the electrons stir and pass to the colder side, thus generating energy. In addition, thermoelectric materials use the material's high temperature to obtain energy or thermal comfort in the environment [13–18].

Pyroelectric is a method of obtaining energy that uses the conversion of a small amount of heat to electricity, often using chemical compositions [13]. The photovoltaic cell acquires energy through solar radiation on the cells' surface, called the photoelectric effect, absorbing photons and releasing electrons, thus generating electric current [14].

## 4. Uses in Civil Engineering

### 4.1. Facades

The facades are the part of the buildings that are exposed to a high incidence of sunlight. In this scope, there are several types of research focused on obtaining energy using this high incidence and natural light. Various types of research use smart windows

and ceramics that capture energy, as well as the fabrics that are used in curtains or flags to generate energy.

Windows are parts of the building that provide a lot of sunlight. Solar cells in window glass use a bifacial light collection technology, an effective strategy to increase the production of energy using photovoltaic devices. The transparent electrode is essential to determine the efficiency of energy conversion. Bi-tandem carbon quantum dots are used to adapt the transparent CoSe to a bifacial electrode in solar cells sensitized with dye, thus achieving an energy conversion efficiency of 8.54% in the front and an efficiency of 6.55% in the rear in comparison with 7.87% and 5.03% in non-carbon devices [5,17–21].

A new smart blind system uses thin-layer photovoltaic cells attached to the blinds to capture energy (Figure 2). The process occurs because the excess of heat, absorbed by the solar cells, becomes dissipated by a coating with porosity at a temperature 9% higher [15]. Moreover, a transparent device has been developed to collect energy from both sides of the material. The material is a synthesis of transparent ternary alloys and economy (CoM) 0.85 Se (M = Ni, Ru, Fe) through a soft solutions method and used as bifacial materials. The efficiency of 9.16%, 8.09% and 7.58% for frontal irradiation and 3.86%, 3.31% and 3.51% for rear irradiation is shown [18].

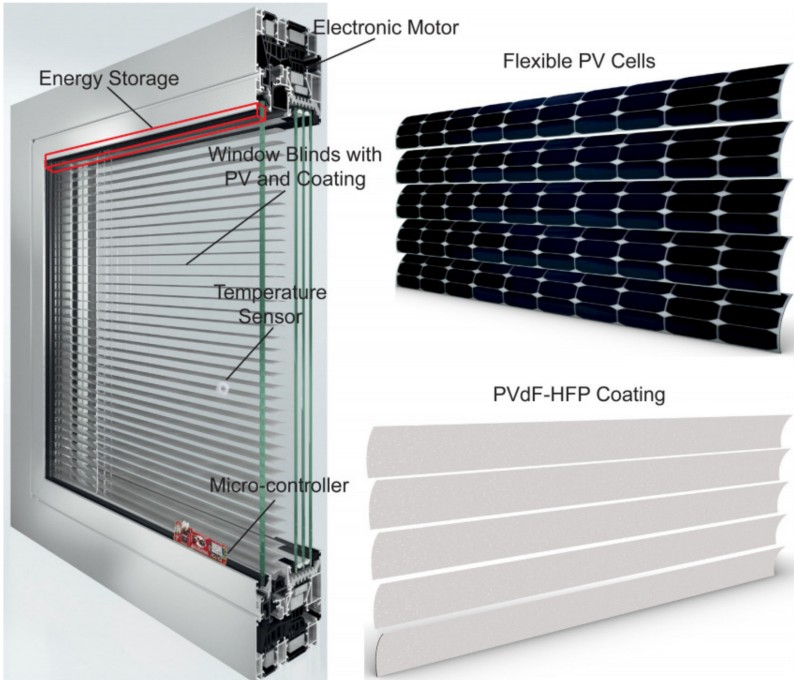

**Figure 2.** Smart window and louver blades [22]. Reprinted from Energy and Buildings, Design and experiment of a sun-powered smart building envelope with automatic control, Qiliang Lin, Yanchu Zhang, Arnaud Van Mieghem, Yi-Chung Chen, Nanfang Yu, Yuan Yang, Huiming Yin, *223*, 110173, Copyright 2020, with permission from Elsevier.

A smart window device with nanofabricated material which comprises a semitransparent perovskite solar cell and an aperiodic multi-layer nanophotonic coating, with different angles from large to large (0°, 15°, 30°, 45° and 60°) was also used. Based on the data analyzed, the calculations were performed on EnergyPlus, projecting an annual energy savings of 13% in a 2-story residential building located in Arizona [17].

A new window was also designed using glass in prismatic gratings filled with water to absorb solar radiation to capture energy. The spectral model has seven bands to investigate solar radiation under various solar conditions and in different locations. The shutter has a dual purpose: energy capture and natural lighting. Using calculations, the device obtained an absorption of 87–88% of the energy of ultraviolet radiation and 81–82% of infrared

radiation and transmitting about 76% of visible light. The total efficiency of solar energy absorption reaches 53–56% [18].

In an existing 20-story commercial building in Lahore, Pakistan, where the climate is hot and humid, origami-shaped glass facades have been placed and are able to control heat loss and provide access to daylight by continuously adapting to varying weather conditions. The origami shape of this facade allows the plant to intercept additional sunlight and align itself to increase the light received, varying its position according to changing sun angles. The shading devices of the proposed facade fold into horizontal and vertical positions, providing solar shading throughout the day. With the origami shaped glass facade in Pakistan, the existing energy load of the building can be reduced by up to 32% when compared to the previous facade, thus being a great option to decrease the energy costs of the building and to mitigate the $CO_2$ incidence in the energy generation [23].

The facades produce energy from microalgae that are grown inside glass elements. The building is almost entirely covered with green, shutter-like glass modules where the green microalgae are grown. This facade cannot only be used to produce energy but can also control light and provide shade. The BIQ House in Hamburg, for example, which contains the world's first "bioreactive" facade known as SolarLeaf, can generate heat, biofuel and shade, as well as decrease $CO_2$ emissions. It is the first building to have a facade with bioreactors [24,25]

The BIQ facade in Hamburg contains glass slides with algae immersed in water. When the algae reproduce, they emit heat, thus heating the water. This water then runs through the pipes heating environments and the water used in the bathrooms and the kitchen. The algae are removed weekly and then taken to places where they are processed to produce fuels such as methane and hydrogen [24,25].

The Hanwa Group manufactures solar panels and other sustainable technologies. It has built a tall tower facade that provides a greater amount of electricity to the building. The building has not only one of the largest solar photovoltaic facades in the world but also an illuminated LED system that makes an attractive animated display at night. In addition, in Portugal, the Solar XXI building, a low-energy office building, uses active and passive solar strategies that have been applied to reduce energy use [26,27].

A team of researchers from the Fraunhofer Center for Photovoltaic Technology, CSP, in the German city of Halle, together with architects from the Leipzig University of Applied Sciences (HTWK Leipzig, Germany) presented a solar facade in the SOLAR.SHELL project. According to experiments, the photovoltaic elements integrated into this facade provide up to 50 percent more solar energy than modules mounted perpendicularly on building facades. The German researchers analyzed how photovoltaic elements can be best mounted on this type of carbon concrete facade, i.e., how to achieve the best result by combining this innovative concrete with solar energy production [28].

Ceramics is another material used in most of the facades which are highly exposed to the sun. TiO 3-based ceramics (Bi 0.5 Na 0.5) is one of the most attractive pyroelectrics, lead-free, due to its high polarization. The reason for carrying out the high energy harvest is due to the addition of Aluminum Nitride (AlN), which can decrease the matrix conductivity, thus resulting in low resistance to rupture (BDS) that allows the application of high electric fields in the polarization, which varies with temperature change [29].

The elements O, Na, Bl and Ti are contained equally in all the materials. In contrast, the Al is concentrated in the material's external parts, consistent with the scattering diffraction mapping. The microstructure proved to be dense and compact; when the AlN content increases by 0.25% by weight, the materials become denser with a grain size increase. When increasing the weight of AlN from 0% to 25%, there is an increase in the polarization saturation from 32 $\mu C/cm^2$ to 41 $\mu C/cm^2$, as well as an increase in the low breaking strength (BDS) of 160 kV/cm to 260 kV/cm, being beneficial for obtaining a high energy capture performance with a greater electric field. The data observed showed an increase of 0.3% in AlN results with a decrease in BDS, thus obtaining a value of 190 kV/cm; this is due to a reduction in density [29].

The capture of thermal energy with high-performance new models of pyroelectric ceramics has heat transfer and high-temperature variation. This was verified with a hexagonal boron nitride (hBN) pyroelectric ceramic with a 0.1% weight of PMN-POM-PZT with ceramic matrix. To obtain a more. detailed comparison of energy consumption, samples were made in cylinders with the same dimension 310 μm thick and 8.5 mm in diameter. In addition, 0.1% of hBN was derived in one sample, and 0.3% hBN in another sample. The hot and cold flow temperature was monitored simultaneously with two regulated type K thermocouples with fan and heat sink. The heating process took place at a temperature of 600 °C for 25 min [29].

The temperature behavior of the relative permissiveness for each addition increased to 213 °C and then decreased with increasing temperature, this temperature being comparable to a transition phase. The actual thermal conductivity (K) occurs with an increase in temperature for each sample. In the sample with the addition of 0.1% hBN by weight, the value of the cyclic temperature changes and the rate of temperature change (dT/dt) increased from 3.19 °C/s to 3.45 °C/s (Figure 3). Furthermore, the 0.1% hBN value showed better results in the saturation and the remaining polarization. Compared to pure ceramic, the 0.1% hBN analysis indicated a 65.6% increase in maximum output power. The ceramic with an addition of 0.1% of hBN by weight has better ferroelectric and pyroelectric properties and better thermal conductivity than ceramic without addition. It shows that the addition can cause a better transfer of heat to capture pyroelectric energy [29].

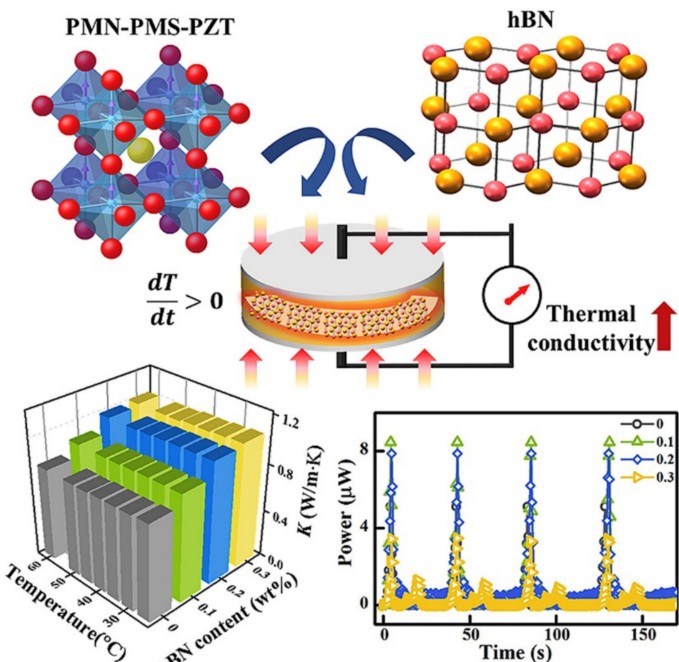

**Figure 3.** Harvesting of thermal energy using pyroelectric ceramics with the addition of hBN with improved heat transfer [29]. Reprinted from Nano Energy, Hexagonal boron nitride nanosheets doped pyroelectric ceramic composite for high-performance thermal energy harvesting, Qingping Wang, Chris R. Bowen, Rhodri Lewis, Jun Chen, Wen Lei, Haibo Zhang, Ming-Yu Li, Shenglin Jiang, *60*, 144–152, Copyright 2019, with permission from Elsevier.

The performance of dielectric absorbers exploiting different coatings consisting of different transparent matrices (SiO$_2$, AlN, SiC) and metallic inclusions, Cu and Ni, combined with selective filters high-temperature applications which are necessary for solar energy. An analysis of coatings for a temperature range between 400 °C and 1000 °C, full recipe at an operating temperature of 850 °C. The best results are coating with copper nanoparticles in a low index dielectric matrix, SiO$_2$ [30].

In addition to ceramics, paints are ubiquitous on facades and can be a great source of energy. The new paints exposed on the surface are composed of nanocomposite films of lead and magnesium titanate (PMN-PT) to capture pyroelectric and piezoelectric energy. The films were manufactured using the conventional brush painting technique to provide low-cost benefits, thus creating multifunctional films. The increase in the films' dielectric constant occurs with the rise in the percentage of PMN-PT in the material. It was analyzed that the voltage and the power increase with the application of thermal variation [31].

In addition to ceramics and windows, many researchers bet on capturing thermal and solar energy through fabrics, which can be used both in curtains and flags. This material is extremely exposed to the sun. A new device that captures wind and solar energy through inverted flags was developed. The method consists of piezoelectric strips and photovoltaic cell strips, both flexible, thus obtaining solar and wind energy. The flags were made with materials existing in the market, without being manufactured, and for that reason, they present some geometric and mechanical limitations [32].

The tests were carried out in a wind tunnel, with a concave circular shape with an orthogonal section; the lighting was composed of two LED lamps placed at a 45° angle at the rear of the flag. The results show that the piezo strips act as cushioning, and the part of solar panels acts as an additional mass of the flags. The analysis shows that since solar panels were slightly less flexible than piezoelectric strips, solar collectors restricted the deformation of piezoelectric materials if they were located near the flag's rear edge. Even with these observations, the new piezo/solar energy collection device may generate 3 to 4 mW of energy, thus being a viable concept and anticipating that, in a real application, several inverted flags adequately adjusted to obtain the desired performance at all wind speeds [32].

In another analysis, a numerical model resulted in a calculation to obtain solar thermal collectors' performance on textiles with different layers. The model presented is based on the numerical resolution of the different elements using an algorithm capable of using each component's different simulation levels. With the model, airflow temperature profiles and useful heat gain were calculated [33].

This system of facade transfers heat through a panel on the exterior wall to a pipe that runs between the building and the panel. This water, as it passes through the pipe, is heated and then stored for use in the shower and kitchen. The heat pipe system provided a greater amount of heat than the other passive options in all climates, with the most advantageous performances in cold and overcast weather. In addition, this type of facade helps with energy waste, because during summer, the solar radiation reaching the vertical surfaces is reduced by containing the pipes, thus avoiding overproduction of thermal energy since the building material does not overheat, and no methods for cooling the room are needed [34,35].

The market for solar thermal collectors is increasing, especially in countries such as Germany and Austria, where appropriate incentives have permitted the development of energy-generating building envelopes [34]. In addition to these countries, Portugal has developed an innovative technology, SENERGY FORCE, which has developed panels that heat water and also provide ventilation and air conditioning, only using the energy of the sun [35].

The project developed by the Portuguese company presents a piping system inside the panel for the circulation of water. When the panel heats up due to the action of the sun's rays, this heat is transferred to the water. In addition, through convection, the heat from the panel enters the building, keeping it warm even during the winter. In addition, it promotes the natural renewal of the air in the environment [36].

In addition to the water collection systems on the facades, there are also integrated photovoltaic/thermal systems (BIPV/T) in the buildings. In these systems, in addition to on-site energy generation, heat recovery also occurs mainly through active cooling of the photovoltaic surface, using open-circuit air [37]. According to some authors [9,38–40],

BIPV/T provides an increase in electricity production, keeping the photovoltaic cell at considerably lower operating temperatures throughout the year.

*4.2. Roofs*

The buildings' roofs also receive a high incidence of sunlight, thus generating energy since they are often not used by the owners. Thus, several studies address techniques for obtaining energy in roofs from photovoltaic cells, TEGs and materials with phase change in gutters.

The use of photovoltaic panels in the building's external area takes advantage of the buildings' innovative planning and configuration so that the performance evaluation is quite beneficial in all the exterior urban areas. The roofs have obtained benefits in cooling the building since the solar panels protect the building. Furthermore, the planning for the deployment of photovoltaic panels in outdoor locations with greater solar incidence reached a high value in obtaining energy, reducing the entire building's net energy costs and indicating an equivalent reduction in $CO_2$ emissions [5,41].

The use of a material to harvest solar energy using photovoltaics or photoelectrochemical due to the material's efficiency, $Cu_2SrSnS_4$ (CSTS), can absorb light. The material was a mixed solution of copper, tin and sulfur with a solution containing strontium. The results show that the film annealed at 600 °C revealed better crystallization; however, for the control of the secondary phases, they were prepared with different atomic values of the strontium/foreign ratio, obtaining a rate of 1.15, 1.30 and 1.45, and thus having the ability to convert photoelectricity [42].

In the north of Taiwan, the energy harvesting analysis from different photovoltaic sources (PV) occurs. The experiments, in the north of Taiwan, were applied with single-axis photovoltaic systems (SASTT), a dual-axis solar tracking (DASTT) and a fixed photovoltaic (FT) system; all systems with a power of 3.68 kW at different inclination angles [43].

A theoretical and an experimental method were used to obtain the analysis of energy harvesting. In the theoretical module, a formula was developed to calculate the inclination angle, the relationship between the sun's rays' angle and each system's electrical parameters. In the experimental model, the energy capture, with an inclination angle of 23.5° from the FT system and the SASTT system, was compared over a total of 6 months. Then, the FT system was compared to two angles of inclination 0° and 23.5° for one year. The experiment was installed on the roof of a factory [43].

In the first experiment, a value of 15.1% of the SASTT was found with the FT system, both with an angle of 23.5° for six months. In the theoretical simulation calculations, 16.1% of the SASTT was obtained with the FT system. Thus, less than 1.2% is observed between the experimental and theoretical part [43].

In the second experiment, the FT system's comparison at 23.5° and 0° angles did not obtain such an explicit increase, acquiring a greater value at the 23.5° angle. The results of the theoretical tests also did not show a significant difference. This is because the sun in Taiwan radiates directly to the ground only in summer, so that over a year of the experiment, it did not vary much [43].

The harvesting of solar energy from ferroelectric materials obtains a high production of photovoltaic energy, acquiring better results than conventional photovoltaic mechanisms. The analyzes show that the $BiFeO_3$ or BFO, as a film in ferroelectric-photovoltaic applications, obtained excellent results in power generation [44].

The temperature difference at the bottom of the photovoltaic panel, where the temperature reaches 70 °C and the ambient air reaches 30 °C. This difference can be used to obtain energy through TEGS. The energy captured by TEGS was 3.3 V with an intensification of around 220 mV obtained with a difference in average monthly temperature of 15 °C. Furthermore, it has been demonstrated that 528 mV can be collected every 30 ms [45].

In addition, the analysis of the temperature difference of the TEGS, using the materials, can be obtained with the hot cell in contact with the black iron roof, and the cold part which is in the shade [33]. In turn, solar energy can be obtained using photovoltaic plates and

thermoelectric sensors clicked on the roof of buildings, thus obtaining a greater use of solar incidence to capture energy [46,47].

The collection of ambient solar energy to charge wireless sensor networks' batteries maximizes the wireless sensor network's service life using the collection of solar energy. The simulators show that the wireless sensor network's lifespan at 25% of the duty cycle is 115.75 days longer than just 5.75 days of solar energy data collection. The use of solar energy capture can benefit an energy restriction project implemented for intelligent monitoring. With this, this network's useful life can be increased for an infinite time using solar energy capture [48].

The latent thermal energy storage method uses phase change material (PCM) at the solar plant where the concentrated parabolic gutter is located. Four different types were used (H250, NaNO$_3$, KNO$_3$ and KOH). The analysis of the use of NaNO$_3$ in the latent thermal energy storage system presented the best option with a solar fraction of 34.14% among the others analyzed. As a result, the solar fraction increased by 90.5% compared to the plant without the system [49].

### 4.3. Structural Materials

The constructions are made of different materials, having different functions in structures. Many of these constructions are in direct contact with the sun's rays, and the materials can bring more advantages to the building with the capture of energy through the high temperatures reached by the sun. With this, several authors approach new materials and additives in the components used in civil engineering that generate solar and thermal energy.

A block of bricks captures energy using electrical generators (TEG) and phase change materials (Figure 4). Power generation is supplied to the block using the residual heat accumulated on the building's external wall surface. The average amount of electricity generated is approximately 0.1 Wh in one block [37]. In addition, there are Fresnel lenses that concentrate the sun's rays on the hot part of thermoelectric generators based on bismuth telluride. The temperature obtained on the hot side was 140 °C, and on the cold side, it was 44 °C, acquiring a value of 0.615 V and 84.9 mA with the temperature difference of a single TEG [50].

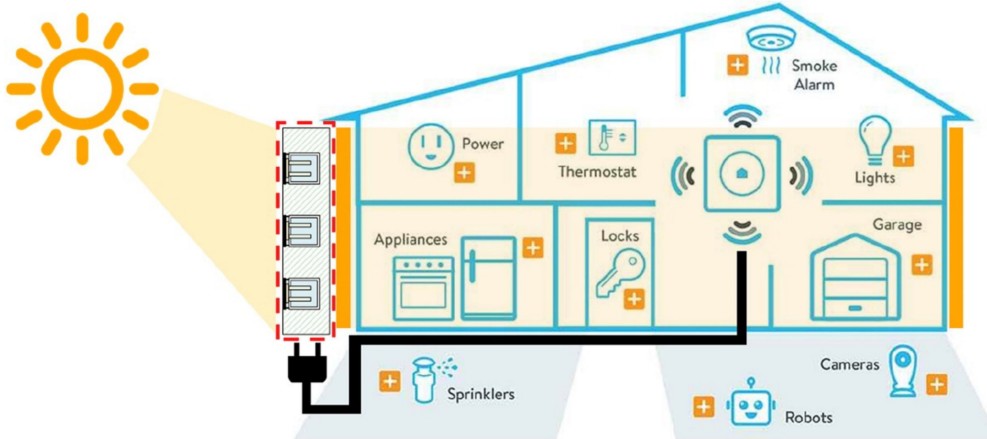

**Figure 4.** Application of the TEG-PCM brick wall [51]. Reprinted from Renewable and Sustainable Energy Reviews, Phase change material-integrated thermoelectric energy harvesting block as an independent power source for sensors in buildings, Yoo-Suk Byon, Jae-Weon Jeong, 128, 109921, Copyright 2020, with permission from Elsevier.

Another way of obtaining energy from the buildings' structure is through thermoelectric materials based on cement which converts the heat absorbed by buildings' surfaces in the summer into electrical energy. These materials can improve buildings' internal climate, energy consumption with interior thermal comfort and external energy capture.

These materials can improve buildings' internal atmosphere and energy consumption with thermal comfort inside and energy capture outside [52–54].

The electrical conductivity of composites rises with the increase in the percentage of graphene (PNB). Just as the conductivity also increases with rising temperature. The addition of 20% of GNP to the cement caused a value of 16.2 σ(S/cm) of electrical conductivity—the Seebeck coefficient for all is positive, which confirms that composites are semiconductors. With 15% of GNP, the highest value of the Seebeck coefficient, 34 μVK$^{-1}$, is obtained at a temperature of 70 °C [53].

Additionally, as the carrier density of the samples increases, the Hall coefficient decreases. The best results of mobility and density of carriers were found for the cement sample with 20% of GNP. The values of thermal conductivity increase with increasing GDP content. However, the impact of GDP content is much more significant on electrical conductivity than on thermal conductivity [53].

The thermoelectric properties of cement with the addition of expanded graphite (EGCC), whose compound has a high Seebeck temperature coefficient with a value of 30 °C to 100 °C as well as an electrical conductivity of 24.8 S/cm. However, the increase in the amount of graphene in the sample resulted in a reduction in compressive strength and increased apparent porosity [54].

Using cementitious materials plus electrochemically exfoliated graphene (EEG) presents better microstructure and mechanical properties and presents high workability concerning cement without EEG. The EEG presents a simple and efficient method that allows a uniform dispersion in the cementitious matrix. It does not agglomerate in the cement's alkaline medium, improving its fluidity and workability. The cement's mechanical properties were analyzed; notably, the addition of 0.05% graphene to Portland cement presents a better result [55].

Analysis of traction results showed an increase of about 79%, as well as an 8% increase in compressive strength and 9% in Young's modulus. Furthermore, graphene presents a hydration reaction of calcium silicates and a compact and regular microstructure. With this, the author's results show that the compound with graphene analyzed can drive a new technology for concrete, allowing the design of lighter and more durable concrete structures [55].

The graphene-added cement used in buildings can offer an advantage not only for capturing energy but also for improving the thermal comfort of the internal atmosphere. Thus, graphene material can provide a very beneficial contribution to construction energy collection applications since it is a low ambient temperature gradient and can be used as a clean energy source to improve energy consumption in construction [53,54].

Solar concrete collectors are a very advantageous alternative for obtaining thermal energy for hot water and space heating applications. The materials used, the geometry, the material surface and the fluid flow were used. Solar concrete collectors are simulated using a 2D finite element model with a 56% daily efficiency predicted for a clear day in Ireland (Figures 5 and 6). The collector covered in this research is an alternative with benefits and durability, such as applications in temperatures below 40 °C [56].

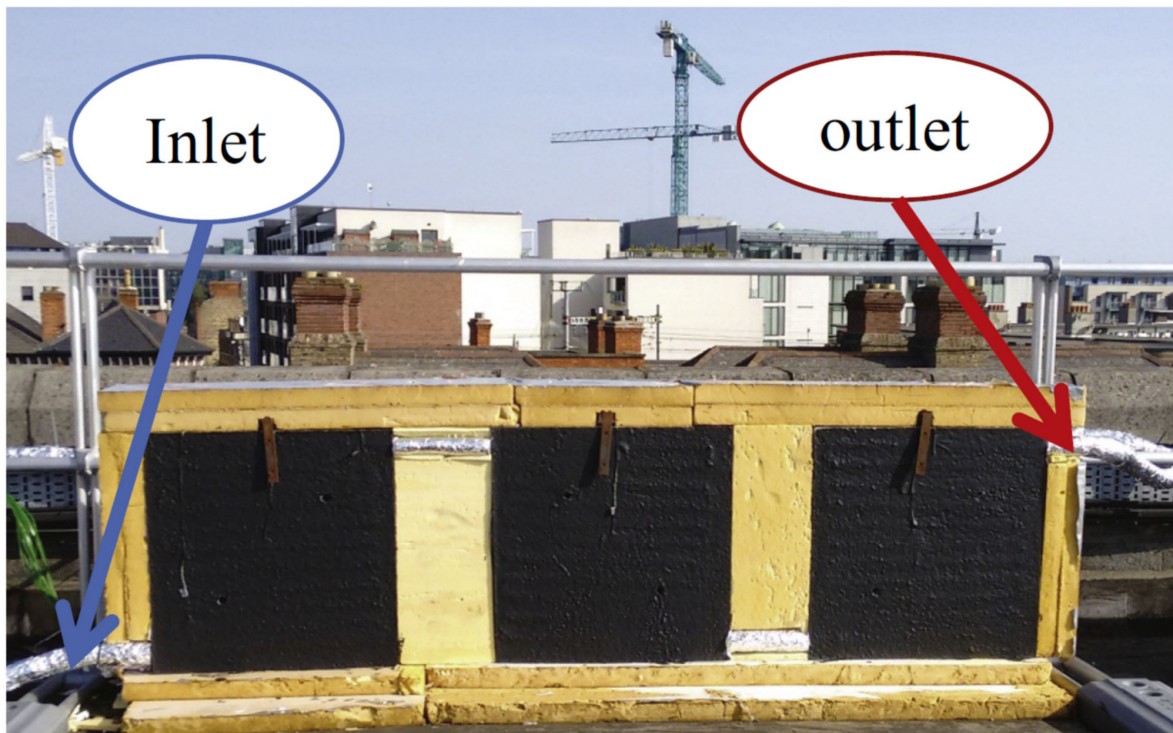

**Figure 5.** Construction of an experimental concrete solar collector in Ireland [56]. Reprinted from Solar Energy, Parametric investigation of concrete solar collectors for façade integration, Richard O'Hegarty, Oliver Kinnane, Sarah J. McCormack, *153*, 396–413, Copyright 2017, with permission from Elsevier.

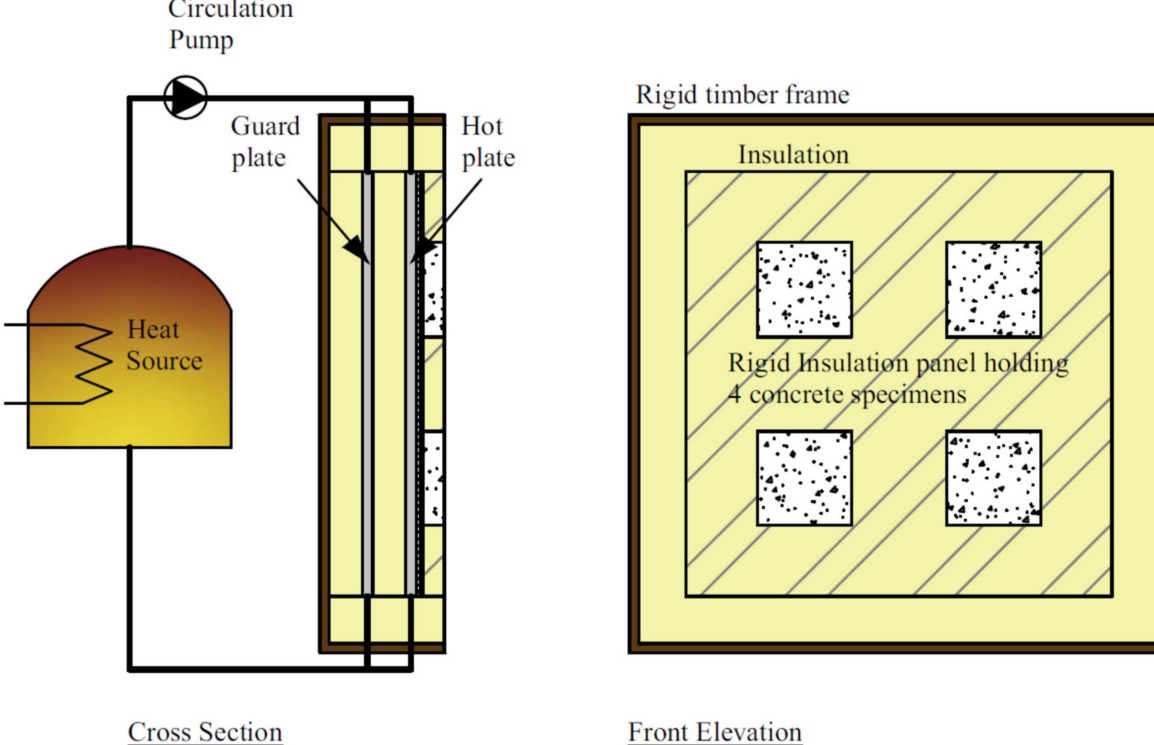

**Figure 6.** Concrete thermal conductivity test schematic [56]. Reprinted from Solar Energy, Parametric investigation of concrete solar collectors for façade integration, Richard O'Hegarty, Oliver Kinnane, Sarah J. McCormack, *153*, 396–413, Copyright 2017, with permission from Elsevier.

The devices that harvest energy from the environment using thermoelectric and phase change materials (PCMs) to supply energy to various electronic devices through heat transfer were designed to improve the performance of the energy capture system. The mechanism was proposed to be built and analyzed in an environment with a temperature difference between 0 °C and 40 °C for three days. The entire system was placed in a simulated environment to respond to changes in temperature [57].

This device is divided into two similar modules, where the only difference is a variety of PCMs with different melting points. The PCM module uses cubic copper foam in paraffin as a thermal insulation container. This storage place is made of transparent plastic and a copper plate; the material is welded to reduce heat loss. For simulation at room temperature, heating a 0 V to 30 V power source was used, and dry ice cubes in the cooling condition [57].

The results presented show that the device for capturing thermoelectric energy in double PCM obtained a load resistance value 35.8% higher than that of simple PCM. In addition, numerical models were performed but showed large deviations in reaction to the experimental ones. The use of the energy harvesting device based on double PCM makes the generation of energy more uniform in situations with great temperature differences, being more efficient for generating energy [57].

Another compound, nanofluids, can be used in the building structure for improved thermal absorption as properties of solar thermal conversation; the fluid contains copper (II) oxide (CuO) and antimony-doped tin oxide (ATO) prepared after surface development of CuO nanoparticles. The results show that the efficiency of solar thermal utilization of the nanofluids of two components together obtained 92.5% when compared to 81.3% of Cuo nanofluids and 80.7% of ATO [58].

However, nanofluids can be used in solar desalination equipment and steam generation equipment to obtain energy [58,59]. In addition, according to the author [48], the comparison of two different types of systems, one for surface absorption and the other for nanofluid, shows that in the absorption of cobalt oxide nanofluid, there was an increase in the average temperature around 23.3 °C. This value is approximately 9.3 °C higher than the surface absorption system.

The phase change through the generation of steam is a source of energy production. The materials used for this purpose were vertically oriented graphene nanofilms and highly porous graphene aerogel to obtain an ultra-fast solar thermal response. A temperature increase of 169.7 in 1 s occurs. This material is heated by the sun's rays and heat transfer; steam occurs with the water heating, thus generating energy [60,61].

Erythritol can cause several disadvantages for energy capture and to solve the supercooling, low conductivity and malabsorption of solar energy from Erythritol, a low-cost commercial metal foam was used through a simple manufacturing process in one rough surface with the use of crushed graphene particles. The results show that the degree of supercooling decreased from 85.8 °C to 26 °C, improved the thermal conductivity from 0.7 W/m K to 4.5 W/m K, and the total absorption of sunlight, indicating a high adaptation for adaptation to solar energy [61].

The mixture of rubber tires and steel fibers presents the thermal behavior of recycled rubber tires reinforced with steel fibers with an excellent performance as solar collectors, showing that the addition of steel fibers can increase the amount of heat absorbed and heating rate of the material. However, adding more than 0.5% of the fiber does not change the temperature. With the addition of 0.5% steel fibers, rubber transmitted heat to the water, reaching 45 °C [62].

### 4.4. Pavements

Pavements are components of a large city made by civil engineering. These structures occupy a large area and, consequently, receive a high incidence of sunlight. Bearing this in mind, several factors were considered by researchers to obtain energy from the pavements. The most used methods in the research are thermoelectric road systems (RTEGS) and

asphalt collectors (ASCs). Several factors approached the methods with new systems and modes of execution to obtain the best harvesting energy results.

The new RTGS system was developed to supply electricity with a difference in temperature and ambient air. With a 1 km long and 10 m wide road, 160 kWh of energy was generated in 8 h of exposure to the sun [63]. Another author also uses TEG to acquire solar energy from the pavement using the temperature difference of the pavement surface and of a lower part of the structure where there was an aluminum bar, obtaining a temperature difference of 20 °C between the materials, thus being sufficient to power an LED light [64]. Furthermore, a thermoelectric system is used on the surface of the pavement and on the ground below the pavement to obtain energy also from the temperature difference [64]. In South Texas, this method was tested using the real dimensions of the highway (10 m wide and 1 km long), and an average of 23.2 kWh of electricity was obtained [65].

The generation of energy using the system that extracts thermoelectric energy from the asphalt pavement can have, according to research, better results by increasing the conversion efficiency of TEGs, defining the ideal system configuration, changing the thermoelectric properties of the pavement and using the point tracking algorithm [53–66]. In addition, a new form of feedback control is required for the system, in which the Air Source Heat Pump (ASHP) ON/OFF connected to the heat storage tank (HST) is activated by comparing the temperature [67]. TEG was installed in water pipes (Figure 7), one heated by the sun's rays and the other cooled by water from a river. The three different temperature types were found: 40.5 °C; 25.5 °C; 11.5 °C, thus obtaining electricity values of 5 W, 2.9 W and 0.9 W [14,68,69].

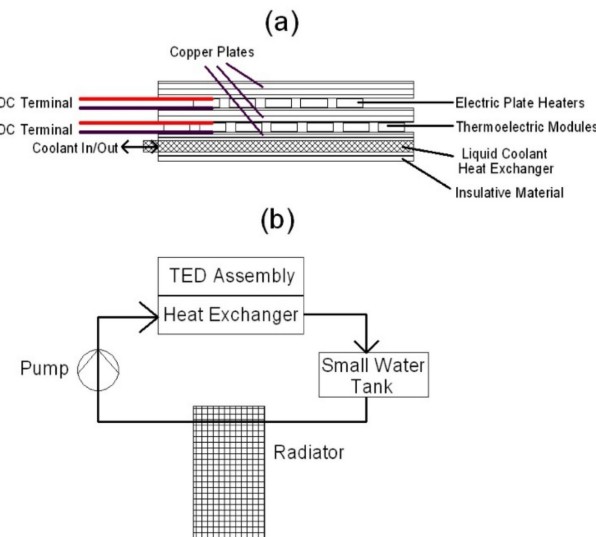

**Figure 7.** Details of the experimental platform (**a**) TED layers, (**b**) liquid cooling system [70]. Reprinted from Control Engineering Practice, Complete implementation of the combined TEG-TEC temperature control and energy harvesting system, Trevor Hocksun Kwan, Xiaofeng Wu, Qinghe Yao, *95*, 104224, Copyright 2019, with permission from Elsevier.

In the networks of fluid tubes incorporated in the pavement structure, the solar asphalt collectors (ASCs), the fluid passes through the duct, which is heated along the way, as there is a transfer of heat from the surface material to the duct and, consequently, for the liquid (Figure 8). The first ASC projects involved metal pipes, but the data was considered favorable for corrosion and leakage. Subsequent projects involved plastic or polyethylene tubes and obtained better performance. The largest increase in temperature between the inlet and outlet was achieved with copper tubes, followed by the radiant PEX-AL with a difference of only 3° C, thus using cheaper materials [14,50,51,71–73].

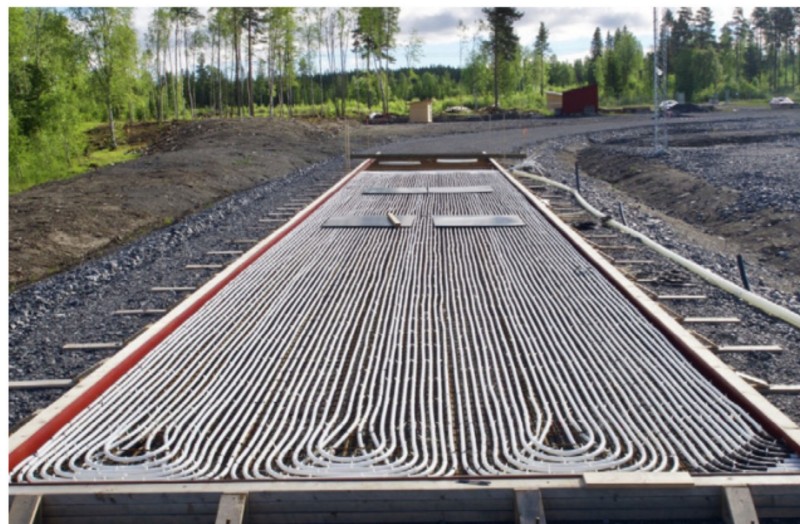

**Figure 8.** Construction of a solar pavement collector near Östersund, Sweden [74]. Reprinted from Applied Energy, A numerical and experimental study of a pavement solar collector for the northern hemisphere, Josef Johnsson, Bijan Adl-Zarrabi, *260*, 114286, Copyright 2019, with permission from Elsevier.

The materials used in paving the new ASC systems have more advantageous criteria for increasing heat transfer with highly conductive materials. However, according to the documents, there was not much change in the results with graphite results. Another solution to increase the systems' efficiency is to paint the floor surface black to increase thermal conductivity [14,75].

The thermoelectric effect through piping on the asphalt pavement is a method that can lower the floor temperature and mitigate the damage caused by high temperature. The floor can also capture clean electrical energy through the piping inserted into the floor and uses the heat collected from the floor to raise the liquid's temperature in the pipe. The energy generation is obtained by the temperature difference between the hot liquid and the tube's cold liquid [7,30,66,71,74].

The photovoltaic materials on the pavement were analyzed and indicate that the application of photovoltaic cells with thinner layers on the pavement presents many challenges since they are difficult to maintain their durability due to high traffic loads and the different climatic conditions floor is exposed to. However, applications of photovoltaic technology protect against noise in the surrounding areas [14,64].

## 5. Advantages and Disadvantages

Each type of solar energy harvest system used in the chosen civil engineering works has its characteristics, with advantages and disadvantages depending on its performance, maintenance and aesthetic potential. The selection of the most suitable system is directly related to the buildings' characteristics and climatic conditions. The Tables 1–4 indicate the advantages and disadvantages according to the place and type of harvest.

**Table 1.** Advantages and disadvantages of facades.

| FACADES | |
|---|---|
| **ADVANTAGES** | Photovoltaic:<br>Use of photovoltaic cells in windows and blinds to obtain greater natural light and solar energy [26–31].<br>Thermal materials:<br>Use of window glass to heat a water pipe to provide solar energy and greater natural lighting [27].<br>Use of fabrics that can be used in lighting locations to generate energy [13,31].<br>Use of anchor paints with components that increase thermal conductivity [30].<br>Pyroelectricity:<br>Use of a new ceramic with components that increase the redemption of solar energy generation. In addition to increasing the breaking strength of the ceramic [27–29]. |
| **DISADVANTAGES** | Photovoltaic:<br>Limited architecture and aesthetics [21–26].<br>Visual pollution [21–26].<br>Thermal materials:<br>Limited architecture and aesthetics [13,22,30,31].<br>Pyroelectricity:<br>Limited architecture and aesthetics [27–29]. |

**Table 2.** Advantages and disadvantages in roofs.

| ROOFS | |
|---|---|
| **ADVANTAGES** | Photovoltaic:<br>The use decreases the internal temperature of the building [32].<br>The CSTS component in the photovoltaic cell increases energy generation [33].<br>The SASTT system showed better power generation than the FT [34].<br>Ferroelectric materials increased the energy generation of photovoltaic cells [35].<br>Thermal materials:<br>The increase in energy generation using $NaNO_3$ material as a phase change in a roof gutter [40].<br>TEGs:<br>Consequently, the temperature difference and energy generation increase through the photovoltaic plates' surface and the shaded part [36,37].<br>The high-temperature difference between the roof and the shaded part [36,37]. |
| **DISADVANTAGES** | Photovoltaic:<br>Visual pollution [32–34]<br>Thermal materials:<br>It was not observed<br>TEGs:<br>It was not observed |

**Table 3.** Advantages and disadvantages in structural materials.

| IN STRUCTURAL MATERIALS | |
|---|---|
| **ADVANTAGES** | Thermal materials:<br>Adding graphene and EGCC to cement increases thermal conductivity [43,44].<br>It improves the internal comfort of the building [43,44].<br>It presents better microstructure and mechanical properties and presents high workability concerning cement without EEG [45].<br>The solar concrete collector achieves a good performance in energy generation and durability. In addition to obtaining water with a different temperature to be used in the building [46].<br>The use of 2 types of phase-changing material makes the generation of energy more uniform with a large temperature difference [44,45]<br>The addition of 0.5% of steel fibers increases energy generation [62].<br>TEGs:<br>Get energy with materials with phase change in bricks.<br>Fresnel lenses concentrate the sun's rays on materials based on bismuth telluride [41] |
| **DISADVANTAGES** | Thermal materials:<br>If excess graphene is added, the porosity increases and the compressive strength decreases [44].<br>TEGs:<br>It was not observed. |

**Table 4.** Advantages and disadvantages in pavements.

| PAVING | |
|---|---|
| **ADVANTAGES** | Photovoltaic:<br>large extension and protection against noise [17]<br>Thermal materials and TEGS:<br>The floor material has high thermal conductivity [48,54].<br>Dark color [31,57,58,62].<br>High conductivity for water in the pipe [75].<br>PEX-AL tubing performs better [17,42,58–63]. |
| **DISADVANTAGES** | Photovoltaic:<br>Low durability due to high traffic loads [18].<br>Thermal materials:<br>Iron pipe showed corrosion and leakage [17,42,59–63].<br>TEGs:<br>It was not observed. |

## 6. Research Needs

As it has been analyzed in recent years, there has been an increase in the number of articles dealing with the capture of renewable energy, thus realizing that reducing pollutants in the environment for energy generation is becoming increasingly more significant. The studies also analyzed studies of different methods, therefore having different options for obtaining energy according to the construction to be carried out. According to another source of information [76], according to the assessment carried out in Portugal, there was an increase in renewable energy generation in 2020 in Portugal. However, the percentage of solar energy is small (Figure 9).

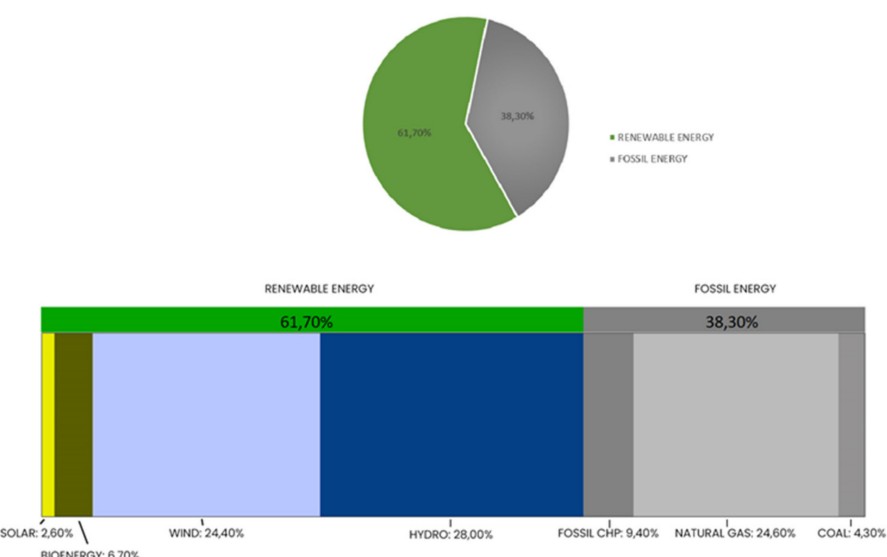

**Figure 9.** Electricity Generation by Energy Sources in Mainland Portugal-adapted from [76].

Therefore, there is a need for further research in the field of harvesting solar energy. According to the analysis of this research, the most studied and relevant method was the use of materials to increase thermal conductivity in all types of energy production analyzed in this study, recognizing that the best knowledge of the materials can provide a better performance of energy. This indicates a need for further studies on the different types of materials, mainly the analysis of waste, to improve solar energy capture.

## 7. Conclusions

The sole purpose of writing this review, which addresses the harvest of solar and thermal energy, is to provide knowledge of existing methods for use in various civil engineering constructions. Our bibliographic study explicitly focused on five types of energy capture: thermoelectric generator, solar asphalt collectors, photovoltaic cells, thermal and pyroelectric materials, divided into local frameworks where they are executed: on facades, roofs and structural materials. The following inferences can be made:

- Different methods of harvesting solar energy and thermal materials using Thermo-electric Generator (TEG), solar asphalt collectors, photovoltaic plates, pyroelectric and thermal materials were obtained through numerical simulations and/or experimental tests in climatic conditions as well as bibliographic articles.
- The number of articles on the topic found on the Science Direct database, between 2017 and 2020, was not as expected. This means that in the last four years, little attention has been given to methods of solar capture and thermal energy to use in civil engineering. The piezoelectric method of capturing energy is much more studied, according to the analyzed articles.

- It was also found that there was an increase in the number of articles on the topic of energy harvest, renewable energy and, consequently, also an increase in the number of articles dealing with solar and thermal energy, but still in a small percentage.
- Studies with these methods are of great importance, and research on renewable energies in different parts of the world is increasingly common. However, studies on energy generation from the sun and temperature are rare, which can be considered quite peculiar because the sun and high temperatures exist worldwide. These methods do not emit pollutants and ensure savings for the population if used in engineering projects, for example.
- With the large amounts of pollutant emissions emitted to the environment, it is very important to study alternative renewable energy generation methods, such as piezo-electric, solar, thermal, marine and wind power. However, similar to many authors mentioned in this document, it is extremely beneficial to develop new materials that provide energy for construction, as shown by the great number of thermal materials' advantages for all energy capture types addressed in this study.

**Author Contributions:** Supervision, J.C.-G.; Writing—Original draft, L.S.; Writing—Review & editing, J.C.-G. All authors have read and agreed to the published version of the manuscript.

**Funding:** This work was also partially financed by Portuguese national funds through FCT-Foundation for Science and Technology, IP, within the research unit C–MADE, Centre of Materials and Building Technologies (CIVE–Central Covilhã–4082), University of Beira Interior, Portugal.

**Data Availability Statement:** Not applicable.

**Conflicts of Interest:** The authors declare no conflict of interest.

## Abbreviations

**ASCs**—Solar Asphalt Collectors
**AC/DC**—Current inverter
**ASHP**—Air Source Heat Pump
**ATO**—Antimony-doped tin oxide
**AlN**—Aluminum Nitride
**Al**—Aluminum
**BDS**—Low burst strength
**BFO** or **BiFeO$_3$**—Bismuth ferrite
**Bi**—Bismuth
**°C**—Degrees Celsius
**CO$_2$**—Carbon dioxide
**CuO**–Copper oxide
**Cu**—Copper
**CSTS**—the theoretical calculation Cu$_2$SrSnS$_4$
**CoSe**—Cobalt Selenide
**CoM**—Synthesis of transparent and economic ternary alloys
**DASTT**—Dual-axis solar tracking
**EEG**—Electrochemically exfoliated graphene
**EGCC**—Cement with the addition of expanded graphite
**Fe**—Iron
**FT**—Fixed-type photovoltaic
**GNP**—Graphene nanoplatelets
**HST**—Heat storage tank

**K**—Thermal conductivity
**LED**—Light Emitting Diode
**Ni**—Nickel
**Na**—Sodium
**O**—Oxygen
**PMN-PT**—Lead magnesium lead titanate
**PMN-POM-PZT**—lead-antimony magnesium niobate lead-manganese-lead zirconate titanate
**PS**—Saturation polarization
**Pr**—Remaining polarization
**PVDF**—Polyvinylidene Fluoride
**PV**—Different photovoltaic
**PNB**—Cement-based composites with graphene nanoplate
**PCMs**—phase change materials
**RTEGS**—Road thermoelectric generator system
**Ru**—Ruthenium
**S/cm**—Siemens/centimeter
**SHTE-AP**—Thermoelectric energy from asphalt pavement
**S**—Sulfur
**SASTT**—Single-axis solar tracking
**SiO$_2$**—Silicon dioxide
**SiC**—Silicon carbide
**Ti**—Titanium
**TEG**—Thermoelectric Generator
**TEG-ASG**—Thermoelectric generator system in solar asphalt collectors

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
