# Peer review of "Review of Energy Harvesting for Buildings Based on Solar Energy and Thermal Materials"

_2673-4109, doi:10.3390/civileng2040046_

Round 1

Reviewer 1 Report

Please see the attached PDF.

Reviewer 2 Report

I appreciate the authors' effort to write a comprehensive literature review in the area of energy harvesting for building heating/cooling and electricity generation. However, a lot of literature in this area is missing. A number of key literature in the area of building integrated photovoltaic thermal systems (BIPV/T) is missing. These are key documents on energy harvesting. In fact, in my humble opinion the phrase “energy harvesting” is not sufficient to search and write literature review. 

Additional comments

Line 16: “MDIP” should this be MDPI?

Line 82-87: I am confused by the following statements (see below). Are you saying that phase 1 is just to set intention and phase to is searching for the phrase “energy harvesting”? Why do you call these steps phases?

In phase 1, the research intention was set and for the articles studied, it was to analyze solar energy capture. The keyword defined to research the databases was "Energy harvesting." The period established for the research was between 2017 and 2020, as it is a new subject where new technologies are essential. In phase 2, the platform was searched with the keyword "energy harvesting", and the documents for each proposed year were analyzed individually.”

I am sorry but I do not see why you use the word “phase”. There is nothing new here. It is just literature review and there is no merit in dividing the task in phases and talking about them. Instead, the authors needed to focus on the literature review itself.

Round 2

Reviewer 1 Report

Overall, the authors have addressed my comments in this revision. Table 1 still spans multiple pages so might be hard to read.

Reviewer 2 Report

The Authors have addressed my concerns. I recommend acceptance of the manuscript.